# The Effect of Resveratrol or Curcumin on Head and Neck Cancer Cells Sensitivity to the Cytotoxic Effects of Cisplatin

**DOI:** 10.3390/nu12092596

**Published:** 2020-08-26

**Authors:** Marinela Bostan, Georgiana Gabriela Petrică-Matei, Nicoleta Radu, Razvan Hainarosie, Cristian Dragos Stefanescu, Carmen Cristina Diaconu, Viviana Roman

**Affiliations:** 1Center of Immunology, Stefan S. Nicolau’ Institute of Virology, 030304 Bucharest, Romania; marinela.bostan@virology.ro; 2Department of Immunology, Victor Babeș’ National Institute of Pathology, 050096 Bucharest, Romania; 3Personal Genetics-Medical Genetics Center, Department of Cytogenetics, 010987 Bucharest, Romania; gabryela.matei@gmail.com; 4Department of Biotechnology, University of Agronomic Sciences and Veterinary Medicine of Bucharest, 011464 Βucharest, Romania; nicolbiotec@yahoo.com; 5Biotechnology Department, National Institute for Chemistry and Petrochemistry R&D of Bucharest, 060021 Bucharest, Romania; 6Otorhinolaryngology and Head and Neck Surgery Department-University of Medicine and Pharmacy “Carol Davila”, 020021 Bucharest, Romania; razvan@riaclinic.com (R.H.); cristiandragos@hotmail.com (C.D.S.); 7Department of Cellular and Molecular Pathology, Stefan S. Nicolau Institute of Virology, 030304 Bucharest, Romania

**Keywords:** resveratrol, curcumin, cisplatin, head and neck cancer, cell cycle, apoptosis

## Abstract

Natural compounds can modulate all three major phases of carcinogenesis. The role of the natural compounds such as resveratrol (RSV) and curcumin (CRM) in modulation of anticancer potential of platinum-based drugs (CisPt) is still a topic of considerable debate. In order to enhance head and neck cancer (HNSCC) cells’ sensitivity to the cytotoxic effects of CisPt combined treatments with RSV or CRM were used. The study aim was to evaluate how the RSV or CRM associated to CisPt treatment modulated some cellular processes such as proliferation, P21 gene expression, apoptotic process, and cell cycle development in HNSCC tumor cell line (PE/CA-PJ49) compared to a normal cell line (HUVEC). The results showed that RSV or CRM treatment affected the viability of tumor cells more than normal cells. These natural compounds act against proliferation and sustain the effects of cisplatin by cell cycle arrest, induction of apoptosis and amplification of P21 expression in tumor cells. In conclusion, using RSV or CRM as adjuvants in CisPt therapy might have a beneficial effect by supporting the effects induced by CisPt.

## 1. Introduction

The heterogeneous nature, at the molecular level, of many types of cancer has hindered both the identification of specific targets and development of efficient therapeutic strategies against tumours. The most common types of treatment for cancer include surgery, radiation and chemotherapy, which can be used either alone or in combination. But these approaches are associated with significant morbidity and a dramatic reduction in quality of life. Head and neck squamous cell carcinomas (HNSCCs) affect different regions of the head and neck, including the tongue, pharynx, larynx, the nasal cavity and salivary glands. The HNSCC is one of the most aggressive cancers, with a complex etiology and pathophysiology [1,2] and these aspects make finding an optimal treatment strategy relatively difficult. Moreover, regardless of the specificity and efficiency of monotherapy, it is difficult to obtain optimal cytotoxic effects on cancer cells because of their molecular adaptability [3]. The poor prognosis of HNSC is due to the occurrence of recurrence or metastasis in many patients after radiation or chemotherapy. Cisplatin (*cis*-diamminedichloroplatinum; CisPt) is the most effective and widely spread chemotherapy choice of treatment available for patients with recurrent and metastatic HNSCC [4]. Unfortunately, there is a high rate of clinical failure of CisPt due to intrinsic or acquired resistance which may lead to therapy discontinuation [5,6]. Tumor cells’ CisPt resistance could occur by reduction of intracellular CisPt accumulation, changes in DNA repair or in the apoptotic cell death pathways [7,8].

In the past few decades, efforts to improve efficacy of cancer treatment have been largely without success, highlighting the need for finding new solutions such as complementary treatment approaches in order to improve the response to chemotherapy [9]. Many natural herbal compounds have attracted the attention of both researchers and clinicians for their use in preventing or improving the current treatment of various chronic diseases, including different types of cancer [10,11,12,13]. Natural compounds are able to interact with multiple cellular targets associated with tumor growth, drug resistance, and metastasis, simultaneously making them a potential source of synergistic cancer treatments. It is possible that, through their multi-targeting activity, natural compounds have the potential to enhance the efficacy of current cancer treatments or to reduce the treatment resistance. The main aim of cancer treatment is to remove or destroy the tumor cells without killing normal cells. Most of the natural compounds have less toxicity, a low cost, are associated with limited side effects and many studies described their effects on the process of carcinogenesis. Due to side effects and drug resistance appearance during conventional therapy, it is becoming obvious that natural compounds have potential to demonstrate anticancer activities or can be used as adjuvants in chemotherapy [14,15,16].

Curcumin (CRM) is a natural compound extracted from the rhizome of turmeric (*Curcuma longa* L.) with reported antiproliferative, antitumoral, antioxidant, anti-inflammatory and chemo-preventive properties and no apparent side effects. In some clinical trials [17,18,19] curcumin use showed low toxicity and good tolerability. CRM exerts anti-carcinogenic activity against a wide variety of human cancers by regulation of various signaling pathways involved in tumorigenesis, gene expression, cell cycle regulation and apoptosis. Curcumin can influence the expression of various protein kinases, transcription factors, inflammatory cytokines and other oncogenic proteins [20,21,22,23].

Resveratrol (3,5,4′-trihydroxystilbene,RSV) is a phytoalexin produced by a wide variety of plants, such as grapes, peanuts and mulberries. This natural compound is one of the most studied componds for its anti-cancer properties besides its other biological properties such as anti-diabetic, anti-platelet, cardioprotective, neuroprotective, anti-aging, antioxidant and anti-inflammatory activity [24,25,26]. Resveratrol appears to be an important player in the fight against cancer, as it may influence the mechanisms responsible for inducing the suppression of tumor cell proliferation, as well as the mechanisms involved in sensitization to chemotherapy [27,28,29]. Rigorous control of cell proliferation and differentiation are necessary to ensure the normal growth and development. Any disorder of the cell division pathways leads to the amplification of the cell division process, the formation of tumors and the appearance of the carcinogenesis process. The carcinogenesis of HNSCC is characterized by multiple events such as activation or suppression of tumour suppressor genes, cell cycle phases disruption, increasing of cell proliferation associated with the decreasing of the apoptotic process [30]. Tumor cells are able to bypass the control point of cell cycle in G1, do not respond to internal regulation and continue to proliferate. It is possible that there are changes in the other phases of the cell cycle, which could be responsible for generating an exaggerated cell proliferation. The balance between cell growth and cell death is influenced by the various molecule regulators like cyclins, cyclin dependent kinases, oncogenes and tumour suppressor genes [31]. One of gene known as a key regulator of the cell cycle as well as cell death and DNA repair is P21 (WAF1/CIP1) a tumor suppressor gene located on chromosome 6 [32,33]. P21 is a cyclin-dependent kinase inhibitor, which is active in response to cellular and environmental signals to develop tumor suppressor activity. In addition, P21 may act as a key mediator of cell cycle arrest after DNA damage [34]. Many studies suggest that P21 gene by direct association with the promoter region of individual genes or by binding to specific transcription factors/coactivators, contribute to modulation of their activity [35,36]. P21 can exert either positive or negative activities toward a specific cellular response in a context-dependent manner, including the cell type and the source of stress signals. Although abnormal expression of P21 gene has been found in various types of malignancy, current views on the role of P21 as a tumor suppressor or tumor-promoting protein remain ambiguous [37,38,39,40,41]. Our study aimed to define the role of P21 on cell control of the cell cycle progression, programed cell death and response to cisplatin in tumor line PE/CA-PJ49 comparatively with normal cell line HUVEC. Despite invasive treatment protocols that comprise surgical resection of the tumor, radiotherapy, chemotherapy and often in combination, the 5-years survival rate of HNSCC patients remain around 40–50% [42]. New therapy approaches are awaited to reduce toxicities, improve survival rates, and quality of life. Natural compounds could be used as adjuvants in HNSCC therapy, due to their good tolerability and low toxicity, as well as their acceptance as dietary supplements [43]. Moreover, numerous studies have displayed the potential utility of natural compounds against HNSCC [44,45]. Currently, there is a great concern about finding natural compounds to support the effects of conventional therapy used in the treatment of HNSCC. The results of this study will provide additional information about P21 gene or protein expression in response to cisplatin mediated by natural compounds (CRM or RSV). Extensive knowledge regarding the molecular mechanisms of natural compounds induced apoptosis, cell cycle regulation and influence on cisplatin response is indispensable for the development of improved therapeutic strategies. In this study we have analyzed the influence of CRM or RSV treatment on protein (p21) or gene expression in HNSCC line (PE/CA-PJ49) treated or not with CisPt, compared to a normal cell line (HUVEC). In addition, we tried to analyze how the CRM or RSV treatment influences the apoptotic processes and the tumor cells arrest in different phases of the cell cycle and the proliferation activity of tumor cells in response to cisplatin therapy. Data analysis was performed in order to investigate possible correlation between the expression level of p21 and the apoptotic process or cell cycle progression in PE/CA-PJ49 tumor cells treated with CisPt in the presence or absence of CRM or RSV compared to the normal HUVEC cell line. The results of this study suggest the use of natural compounds CRM or RSV, as adjuvants in order to improve the response to CisPt therapy.

## 2. Materials

Dulbecco’s modified Eagle’s medium (DMEM), fetal bovine serum (FBS), glutamine, cisplatin (*cis*-diammineplatinum (II) dichloride, DDP), resveratrol (3,5,4′-trihydroxystilbene), curcumin, 0.9% sodium chloride (NaCl), dimethyl sulfoxide (DMSO), trypsin-EDTA (0.25% trypsin, 0.03% etylenediaminetetraacetic acid), phosphate buffer solution (PBS) and 3-(4,5 dimethylthiazol-2-yl)-2,5 di-phenyltetrazolium bromide (MTT), propidium iodide (PI) stock solution (4 mg/mL PI in PBS); and RNase A stock solution (10 mg/mL RNase A) were purchased from Sigma-Aldrich (St. Louis, MO, USA). CellTiter 96^®^ AQueous One Solution Cell Proliferation Assay (G3580) was from Promega (Madison, WI, USA). RIPA buffer was obtained from Pierce (Rockford, lL, USA), protease inhibitor cocktail and phosphatase inhibitor were from Roche (Rockford, lL, USA); BCA protein assay kit was from Thermo Scientific (Waltham, MA, USA); DuoSet^®^ IC ELISA Intracellular Human Total p21/CIP1/CDKN1A (DYC1047-2) and stop solution–2 N H_2_SO_4_ were purchased from R&D Systems Inc. (Minneapolis, MI, USA); the Annexin V-FITC kit was from BD Pharmingen (San Jose, CA, USA); TRIzol reagent was from Invitrogen (Carlsbad, CA, USA); 96-well plates, micropipettes, special tips, Eppendorf tubes; High-capacity cDNA Reverse Transcription Kits (Applied Biosystems, Foster City, CA, USA) contained the following components: RT-Buffer 10×, 1 mL; RT-10× random primers, 1 mL; 25× dNTP mix (100 mM); MultiScribe ™ Reverse Transcriptase, 50 U/μL; RNase inhibitor 100 uL; ultrapure water; TagMan validate Hs00355782_m1 gene CDKN1A and Hs02800695_m1 gene HPRT1; QPCR master mix; plates (MicroAmp Fast Optical 96-Well Reaction Plate, 0.1 mL).

## 3. Methods

### 3.1. Cell Lines and Treatment

The cell lines were obtained from European Collection of Authenticated Cell Cultures (ECACC, Salisbury, UK). The squamous carcinoma cell line PE/CA-PJ49 (ECACC cat. no. 0060606) was obtained from a 57-year old male patient with tongue carcinoma. The PE/CA-PJ49 cell line was grown and maintained in DMEM supplemented with 10% FBS, 2 mM glutamine, at 37 °C in 5% CO_2_.

Human Umbilical Vein Endothelial Cells line (HUVEC, ECACC cat. no.06090720) was maintained at 37 °C and 5% CO_2_ in complete endothelial cell growth medium (ECACC cat. no.06091509). The sub-confluent cultures of both cell lines (70–80%) were split 1:4–1:8 (i.e., seeding at 1–3 × 10,000 cells/cm^2^) using trypsin-EDTA (0.25% trypsin, 0.03% EDTA) according to the manufacturer’s instructions [46].

#### 3.1.1. Preparation of Stock Solution

CRM and RSV were dissolved in DMSO to make a stock solution of 10 mM. Cisplatin was prepared as a 10 mM stock in 0.9% sodium chloride (NaCl). The solutions stock were filtered through a 0.22 μm membrane, aliquoted and stored at −20 °C until further use. The cisplatin, curcumin or resveratrol stock was used to obtain the working concentrations related to the treatment by performing dilutions in the culture medium [47].

#### 3.1.2. Cell Cultures Treatment

Cells were cultured in specific cell line medium supplemented with 10% inactivated FBS, 2 mM L-glutamine and incubated at 37 °C and in 5% CO_2_ humid atmosphere for 24 h to achieve around 60–70% confluence, and then treated with various concentrations of CisPt, CRM or RSV for different periods of time (6, 24, 48 h). After treatments, adherent cells were detached from flasks with a trypsin-EDTA solution, washed twice in PBS. PE/CA=PJ49 and HUVEC cells were used for the evaluation of cytotoxicity capacity, apoptosis, distribution of cell cycle phases or for storing as cell pellets at −80 °C for preparation of cell lysates for further use in ELISA or PCR assays. In all experiments described in this study, all untreated cells were designated as control cells. For treated cells, the conditions (dose and treatment duration) and reagents that were used together at specific dose, were indicated in the figure legends. Non-treated cells were used as controls throughout the experiments.

### 3.2. Drug Sensitivity Assay (MTT)

The MTT assay is based on the ability of viable cells to reduce the reagent 3-(4,5-dimethylthiazol-2-yl)-2,5-diphenyltetrazolium bromide to colored formazan compounds [48]. Because the transformation is only possible in viable cells, the amount of blue formazan is proportional to the number of viable cells and thus, a linear dependence exists between cell activity and absorbance. Cells were seeded in triplicate in 96-well plates at a density of 5000 cells/well, incubated at 37 °C for 24 h. Then cells were treated with or without different concentrations of CisPt (1–80 μM) and CRM or RSV (1–160 μM) for the different time period (6, 24, and 48 h). After the incubation period, 20 μL MTT (5 mg/mL in PBS) was added and further incubated for 4 h. The supernatant was removed, and the formazan product was analyzed spectrophotometrically (570 nm) after dissolution in DMSO. Wells without cells serve as a blank, and their absorbance was subtracted from the other results. Untreated cells were used as control and their viability was assumed as 100%. Results were expressed as mean values of three determinations ± standard deviation (SD). Data are presented as percent of cell viability and compared to untreated [Viability% = (T − B)/(U − B) × 100, {where T = absorbance of treated cells; U = absorbance of untreated cells, B = absorbance of blank}]. Once percent viability was obtained, the drug response curve was generated, and inhibitory concentration (IC_50_ and IC_25_) was calculated using GraphPad Prism version 5.01 [49].

In the present study, the degrees of selectivity of the CisPt, RSV or CRM are expressed as a selectivity index (SI): SI=IC_50_ of compound in a normal cell line/IC_50_ of the same compound in cancer cell line, where IC_50_ is the concentration required to kill 50% of the cell population. SI values of more than 2 were considered as indicative of high selectivity [50,51].

### 3.3. Cell Proliferation Assay

In order to determine the effect of CRM or RSV on proliferative activity of cisplatin treated cells was used the Promega CellTiter 96^®^ AQueous One Solution Cell Proliferation Assay, a test that is based on the reduction of yellow MTS tetrazolium salt by the viable cells and generation of colored formazan soluble in the culture medium. The product was spectrophotometrically quantified by measuring the absorbance at λ = 490 nm using a Dynex plate reader (DYNEX Technologies-MRS, Chantilly, VA, USA) [52,53]. Results were expressed as mean values of three determinations ± standard deviation (SD). Untreated cells served as control and considered to have a proliferation index (PI = absorbance of treated cells/absorbance of untreated cells) equal to 1.

### 3.4. Cell Lysates

Cells were treated for 24 h with CisPt, RSV and CRM at the indicated concentrations. After incubation, cells were washed twice with ice-cold PBS, scraped, pelleted and lysed in RIPA buffer (Pierce) supplemented with protease inhibitor cocktail (Roche) and phosphatase inhibitor (Roche) for 30 min on ice. The lysates were transfered into microcentrifuge tubes and centrifuged at 14,000× *g* for 5 min at 4 °C followed by sonication (10–15 s × 3). The supernatants were collected into clean microcentrifuge tubes and stored at −80 °C. Lysates protein concentration were quantified by BCA protein assay kit (Thermo Scientific) and aliquots of 50 μg of total cell lysate were used for ELISA assay [54].

### 3.5. ELISA Assay

DuoSetR IC ELISA-Human Total p21/CIP1/CDKN1A(: DYC1047-2) was purchased from R&D Systems Inc. and contains the basic components required for the development of sandwich ELISAs to measure human p21 protein also known as CIP1 and CDKN1A in cell lysates. An immobilized capture antibody specifically binds human p21. After washing away unbound material, a biotinylated detection antibody specific for human p21 is used to detect the captured protein, utilizing a standard Streptavidin-HRP format [55]. All experiments were performed in triplicates and absorbance was measured at λ = 450 nm using a Dynex plate reader.

### 3.6. Apoptosis Analysis

The apoptosis assay was carried out with the Annexin V-FITC kit using the manufacturer’s (BD Pharmingen) protocol. Treated and untreated 1 × 10^6^ cells/mL were resuspended in cold binding buffer and staining simultaneously with 5 μL FITC-Annexin V (green fluorescence) and 5 μL propidium iodide (PI) in the dark, at room temperature for 15 min. Then, 400μL of Annexin V binding buffer was added and 10,000 cells/sample were acquired using a BD Canto II flow cytometer. The analysis was performed using DIVA 6.2 software in order to discriminate viable cells (FITC^−^PI^−^) from necrotic cells (FITC^+^PI^+^) and early apoptosis (FITC^+^PI^−^) from late apoptosis [56,57].

### 3.7. Cell Cycle Analysis by Flow Cytometry

For analysis of cell-cycle distribution, the cells were collected after treatment, fixed with 70% ethanol for at least one hour at 4 °C. The cells were stained with propidium iodide (PI) an agent which intercalates into the major groove of double-stranded DNA and produces a highly fluorescent adduct that can be excited at 488 nm with a broad emission around 600 nm. Since PI can also bind to double-stranded RNA, it is necessary to treat the cells with RNase for optimal DNA resolution. Cells (1×10^6^ cells/mL) were washed in PBS and centrifuge at 300× *g*, 5 min at 4 °C. The pellet of the cells were incubated for 10 min at 37 °C with 0.5 mg/sample RNase A and then was added 10 μg/sample of PI staining solution to cell pellet, mix well and incubated 10 min at 37 °C. The samples stored at 4 °C until analyzed by flow cytometry. A minimum of 20,000 events for each sample were collected using a FACS CantoII flow cytometer and ModFIT software (BectonDickinson) and used to determine the cell cycle phase distribution after debris exclusion [58,59].

### 3.8. RT-PCR [60,61]

#### 3.8.1. RNA Isolation with TRIzol

In order to isolate of total RNA from treated or untreated cells was used TRIzol reagent (Invitrogen)-a monophasic solution of phenol and guanidine isothiocyanate. During the isolation phases, the TRIzol reagent maintains RNA integrity, while disrupting cells and dissolving cell components. The addition of chloroform followed by centrifugation ensures the separation phase of the solution into an aqueous phase and an organic phase. RNA remains in the aqueous phase and is recovered by isopropanol precipitation. The RNA pellet washed once with 75% ice-cold ethanol and entrifuged at 10,000× *g* for 5 min at 4 °C. The RNApellet is redissolved in nuclease-free water and incubating in a 60 °C water bath for 10 min. The concentration of isolated RNA was assessed using a NanoDrop spectrophotometer (NanoDropTechnologies, Montchanin, DE, USA). RNA purity was determined by A_260_/A_280_ ratios (an A_260_/A_230_ ratio as close as possible to 2 indicates the presence of highly purified RNA).

Reverse transcription of messenger RNA molecules was performed using the High-capacity cDNA Reverse Transcription Kit from Applied Biosystems, using non-specific, randomic primers, following the manufacturer’s instructions (Table 1). To perform cDNA synthesis 1µg of total RNA was used. The obtained cDNA was stored at 4 °C and further used in the amplification reaction Real-Time PCR (RT-PCR) [60,61].

#### 3.8.2. Real-Time PCR

The analysis of the CDKN1A gene (P21) expression level was performed by real time PCR using a ViiA ™ 7 Real-Time PCR System by setting the ABI 7500 Fast program (Applied Biosystems). The reference gene used in the experiments was hypoxanthine ribosyltransferase (HPRT1) because this gene is found in all cell types and has a stable, relatively constant expression regardless of experimental conditions. The reference gene is useful in calibrating and interpreting qRT-PCR. The products obtained after reverse transcription reaction were diluted to a final concentration of 100 ng/uL with DEPC-treated water. The samples were prepared according to the standard protocol kit, in a reaction volume of 20 μL (Table 2):

Each sample was performed in duplicate. Thermal cycling conditions of PCR were as follows: 95 °C for 10 min for amplification activation and 40 cycles at 95 °C for 12 s and 60 °C for 15 s.

The samples were analyzed using the formula 2^−ΔΔCt^. The value obtained indicates how many times the expression of the gene has increased or decreased compared to the control sample (untreated cells):ΔCt1 = Ct (gene of interest) − Ct (reference gene) (treated cells)
ΔCt2 = Ct (gene of interest) − Ct (reference gene) (control cells)
ΔΔCt = ΔCt1 − ΔCt2
gene expression = 2 − ΔΔCt

### 3.9. Statistical Analysis

Data analyses were performed using GraphPad Prism 7 (GraphPad Software Inc., La Jolla, CA, USA). The differences between the treatment and control groups were statistically analysed using unpaired two tailed t-test and one-way ANOVA. Statistical significance was considered at *p* < 0.05.

## 4. Results

### 4.1. Effects of CisPt and/or Natural Compounds RSV, CRM on Cell Viability

The anticancer cytotoxic activity in vitro was evaluated by the MTT assay. The tumor cells (PE/CA-PJ49) and normal cells (HUVEC) were treated for 24 h with different concentrations of CisPt (1, 2.5, 5, 10, 20, 30, 40, 80 and 160 μM) and/or natural compounds CRM or RSV (1, 5, 10, 20, 30, 40, 80 and 160 μM). The most widely used and informative measure of a drug’s efficacy is IC50-the half-maximal inhibitory concentration. IC_50_ was determined at half of the difference between the maximum (plateau) and minimum absorbance values, by plotting the absorbance value at 570 nm (Y axis) versus the concentration of the compound analysed (X axis). The IC50 values were reported as a mean of three independent experiments ± standard deviation (S.D.). In addition, we also determined IC 25 for both CisPt and CRM and ESV in order to choose the optimal working concentrations for these compounds (Table 3). Results showed that in comparison to the control, both CisPt and CRM or RSV caused dose-dependent toxicity. The IC_50_ values (±SEM) for CisPt were 9.72 ± 1.7 μM on PE/CA-PJ49 cells and 20.93 ± 2.1 μM on HUVEC cells (Table 3a). RSV, under our experimental conditions, had an IC_50_ of 46.8 ± 2.6 μM on PE/CA-PJ49 cells and 110.4 ± 8.6 μM on HUVEC cells (Table 3a). CRM treatment for 24 h reduced the cellular viability to an IC_50_ = 16.3 ± 3.4 μM on PE/CA-PJ49 tumor cells and IC50 = 59.3 ± 6.1 μM on normal cells HUVEC (Table 3a).

Selectivity index (SI) values were also calculated for RSV or CRM on both cell lines and compared to those calculated for CisPt. The results are presented in Table 3b. The SI value calculated for RSV (2.36) was close to the SI value for CisPt (2.15), but he highest SI value was obtained for CRM (3.66). Knowing that the greater the SI value is, the more selective it is and SI values less than 2 indicate general toxicity [62], we concluded that compared to CisPt, the common chemotherapy drug, RSV (SI = 2.36), works in a similar manner as CisPt, while CRM exhibits a high degree of cytotoxic selectivity (SI = 3.66).

The aim of our study was to use natural compounds capable of potentiating the cytostatic effects of CisPt without having a toxic effect on their own on normal cells. For this purpose, we also determined IC25 for CisPt, CRM or RSV treatments in order to select the optimal concentrations used in the combined treatment. The optimal time for cell-treatment was 24 h and the used concentrations for the natural compounds were 15 μM for CRM and 40 uM for RSV for treatment of both cell lines as shown in Figure 1 For CisPt treatment, the choice to use 2 concentrations was made (5 μM and 20 μM) in order to determine if the modulatory effects induced by the two natural compounds were dependent by the CisPt dose (Figure 1).

The obtained results showed that the cisplatin treatment affects the viability of both normal HUVEC cells and PE/CA-PJ49 tumor cells. As shown in Figure 1, the viability of tumor cells PE/CA-PJ49 is significantly affected by CisPt treatment 5 μM (** *p* = 0.002) or 20 μM (*** *p* < 0.0001), compared to untreated cells. Analysis of the effect of CisPt treatment on the viability of PE/CA-PJ49 tumor cells showed that the difference between the effect induced by 20 μM CisPt treatment is significantly higher than the one induced by 5 μM CisPt treatment (** *p* = 0.0058).

CisPt treatment of normal human HUVEC cells showed that regardless of the chosen concentration CisPt 5 μM (** *p* = 0.0055) and CisPt 20 μM (*** *p* = 0.0002) the cell viability was significantly reduced compared to untreated cells. In addition, CisPt 20 μM treatment of HUVEC cells led to a decrease cell viability compared to the effect induced by treatment with 5μMCisPt (** *p* = 0.0081). Comparative analysis of the effect of CisPt treatment on the two cell lines led to the following observation-both the low-concentration 5μM CisPt (non-significant, *p* = 0.287) and the high-concentration 20μMCisPt (non-significant, *p* = 0.105) treatment acted similarly on the viability of tumor and normal cells.

The viability of tumor cells PE/CA-PJ49 (*** *p* = 0.0004) or normal HUVEC cells (** *p* = 0.0077) is significantly affected by RSV 40 μM treatment compared to untreated cells (Figure 1). The effect of the 40 μM RSV treatment on cell viability does not significantly differ from the effect induced by 5 μM CisPt on both normal cells (non-significant, *p* = 0.578) and tumor cells viability (non-significant, *p* = 0.0891). When analyzing the effect of the 40 μM RSV treatment on PE/CA-PJ49 tumor cells, compared to the induced effect on normal HUVEC cells, it was observed that RSV reduced the viability of tumor cells more than in the case of normal cells (* *p* < 0.013) (Figure 1).

Treatment with 15 μM CRM reduced tumor cell viability compared to normal cells HUVEC (* *p* < 0.015). The effect induced by CRM treatment on tumor cells PE/CA-PJ49 was not different from the effect induced by treatment with 5 μM CisPt (non-significant, *p* = 0.578), but was different compared to the effects induce by the 20 μM CisPt treatment (** *p* = 0.0077), (Figure 1).

### 4.2. Effects of CisPt and/or Natural Compounds RSVor CRM on the Cell Proliferation Process

PE/CA-PJ49 and HUVEC cells were treated with 5 μM and 20 μM CisPt and/or 40 μM RSV, 15 μM CRM for 24 h. Analysis of the effect induced by the individual treatment with CisPt, RSV or CRM on the proliferation process of the PE/CA-PJ49 tumor cells and of the normal HUVEC cells are shown in Figure 2A,B. Treatment with 20 μM CisPt for 24 h determined a significant decrease of the proliferation process compared to the 5 μM CisPt treatment, both in the case of tumor cells PE/CA-PJ49 (** *p* = 0.0065) and that of normal cells HUVEC (** *p* = 0.0011). Treatment with 40 μM RSV for 24 h affected the proliferation of tumor cells PE/CA-PJ49 much more than the one of normal cells HUVEC (** *p* = 0.0063) (Figure 2A).

Analysis of the effect induced by the combined treatment of 5 μM CisPt + 40 μM RSV on PE/CA-PJ49 tumor cells showed that RSV can amplify the effect induced by lower CisPt concentration by recording a decrease of the IP (5 μM CisPt + 40 μM RSV vs. 5 μM CisPt; * *p* = 0.0145). In case of applying the same treatment on the normal HUVEC cells no significant changes of IP were registered (5 μM CisPt + 40 μM RSV vs. 5 μM CisPt; non-significant, *p* = 0.105). Using a high concentration of CisPt, the proliferative response of tumor cells to the combined treatment of 20 μM CisPt + 40 μM RSV recorded a significant decrease in IP (20 μM CisPt + 40 μM RSV vs. 40 μM RSV; ** *p* = 0.0093). Treatment of normal cells with 20 μM CisPt + 40 μM RSV led to a response similar to that induced by treatment with 20 μM CisPt (non-significant, *p* = 0.53), but there was a significant decrease in IP compared to the response induced by treatment with 40 μM RSV alone (** *p* = 0.005) (Figure 2A). These results show that the effect of treatment with 20 μM CisPt is dominant in the treatment of normal or tumor cells. Treatment with 15 μM CRM for 24 h on tumor cells PE/CA-PJ49 led to a much more intense decrease of the proliferative process compared to the effect induced on normal HUVEC cell proliferation (** *p* = 0.0065) (Figure 2B). Applying the combined 5 μM CisPt + 15 μM CRM treatment to normal cells did not alter the proliferative process compared to the effect induced by individual treatment with 5 μM CisPt (non-significant, *p* = 0.784) or 15 μM CRM (non-significant, *p* = 0.209). IP of tumor cells PE/CA-PJ49 treated with 5 μM CisPt + 15 μM CRM suggested a significant inhibition of the proliferative process compared to the effect induced by treatment with 5 μM CisPt (** *p* = 0.008) or 15 μM CRM (** *p* = 0.0042). The combined treatment of 20 μM CisPt + 15 μM CRM very strongly inhibited the proliferation of tumor cells compared to the effect induced by treatment only with CRM (*** *p* = 0.0006) (Figure 2B). The combined treatment applied to normal cells HUVEC led to a decrease in IP, but the dominant effect on cell proliferation appears to be due to the high concentration of CisPt (20 μM CisPt + 15 μM CRM vs. 15 μM CRM; * *p* = 0.0424). Comparative analysis of the proliferative response of PE/CA-PJ49 tumor cells versus the normal HUVEC cells to the combined CisPt + CRM treatment shown a significantly different cellular behavior (** *p* = 0.004 –5 μM CisPt + 15 μM CRM; ** *p* = 0.0051 – 20 μM CisPt + 15 μM CRM) (Figure 2B).

### 4.3. Modulation of p21 Protein Expression by Natural Compounds and/or Cisplatin Treatment

In cancer development and evolution p21 protein might acts as an oncogene or tumor suppressor and for this reason it could be an important player in processes such as the cancer aggressiveness or the response to chemotherapy [63,64]. In this study, using ELISA assay we analyzed the expression level of p21 in the tumor cells (PE-CA/PJ49) compared to a normal cells (HUVEC). In addition, we looked at how treatment with CisPt, RSV or CRM applied individually or in combination could affect cell p21 protein expression. The data showed that the p21 protein is expressed more (3.4×) in the untreated tumor cells of the PE-CA/PJ49 vs. normal cells of the HUVEC line (625/184 pg/mL). Individually applied treatment with CisPt, 40 µM RSV or 15 µM CRM did not significantly affect the expression of p21 protein (*p* = non-significant) in normal HUVEC cells. Treatment of normal cells with 20 µM CisPt + 40 µM RSV induced an increase of p21 protein expression compared to untreated cells (*p* < 0.029) or to cells treated only with RSV (** *p* < 0.0085). A significant increase in p21 protein expression of HUVEC cells was recorded in the case of the combined treatment 5 µM CisPt + 15 µM CRM compared to untreated cells (** *p* < 0.0048) (Figure 3 and Table 4).

PE/CA-PJ49 tumour cells were treated 24 h with 5 µM CisPt (*** *p* = 0.0009) or 20 µM CisPt (** *p* = 0.0039) showed a significant increase of total p21 protein expression compared to untreated cells (control). 40 µM RSV (**** *p* < 0.0001) applied alone, determined a significant increase of the total p21 protein expression in PE/CA-PJ49 cells versus control (Figure 3). On the contrary, after combined treatment with 5 µM CisPt + 40 µM RSV (* *p* < 0.014) versus treatment with 5 µM CisPt alone the results showed a slight increase in p21 expression of PE/CA-PJ49 cells. As shown in Figure 3 and Table 4 the simultaneous treatment with 20µMCisPt+40µMRSV induced in tumor cells a significant decrease of p21 protein expression comparatively with the effect induced by RSV alone (** *p* = 0.0058). These results support the notion that the dominant effect is attributed to the high concentration of CisPt. Thus we can say that RSV can potentiate the induced effect of 5 µM CisPt but cannot counteract the induced effects of treatment with a much higher concentration of CisPT on p21 protein expression in tumor cells.

PE/CA-PJ49 tumor cells treatment with 15 µM CRM led to a significant increase of the p21 protein expression compared to control (** *p* = 0,0022). Combined treatment 15 μM CRM and 5 μM CisPt (**** *p* = 0.0001) or 20 μM CisPt (**** *p* = 0.0001) led to a much higher amplification of p21 protein expression compared to CisPt only-treated cells (Figure 3 and Table 4). Our data showed that in PE/CA-PJ49 tumor cells the effect induced by the CisPt and CRM was cumulative. In addition, the analysis reveal the potential of CRM to sustain and amplify the effect induced by CisPt treatment, regardless of the CisPt concentration used.

### 4.4. Modulation of P21 Gene Expression by Natural Compounds and/or Cisplatin Treatment

Changes of the P21 gene expression have been found in various types of malignancy including head and neck [65,66,67] but the impact of the P21 level on the disease progression and prognosis remains controversial. Therefore, we performed in vitro experiments to assess how natural compounds (RSV or CRM) modulated the P21 expression in head and neck tumor cells PE/CA-PJ49 versus normal cells HUVEC. Analysis of the expression level of P21 was performed by real-time qPCR assay in order to identify their role in response to chemotherapeutic agent-CisPt. Obtained data from this study may help define how P21 acts in modulating the DNA repair processes of tumor cells, in hope of finding new effective strategies in the treatment of head and neck cancer. Using real-time qPCR assay, we evaluated the P21 gene expression induced by CisPt and/or RSV, CRM treatment of HUVEC and PE/CA-PJ49 tumor cells. The data showed that RSV treatment acts differently on the two cell lines analyzed. RSV induces a slight decrease in P21 gene expression in HUVEC cells and an increase in the PE/CA-PJ49 tumor cells. Treatment with 5 μM CisPt (* *p* = 0.027) or 20 μM CisPt (** *p* = 0.016) induces an antagonist response compared to the effect induced by RSV treatment alone in P21 gene expression of normal HUVEC cells. The results obtained from the combined CisPt + RSV treatment show that RSV did not significantly modify the P21 gene expression generated by CisPt treatment in normal cells. CRM treatment applied alone did not affected P21 expression in HUVEC cells. The combined 5 μM CisPt + 15 μM CRM treatment showed a significant increase in P21 expression in normal cells (** *p* = 0.019 versus 15 μM CRM). On the contrary, 20 μM CisPt + 15 μM CRM treatment of HUVEC cells induced a decrease of P21 gene expression compared with the effect induced by CRM alone (* *p* = 0.021) or 20 μM CisPt (* *p* = 0.013) (Figure 4 and Table 5).

RSV alone induced a significant increase of P21 expression (** *p* = 0.006) while 5 μM CisPt (*p* = non-significant) applied individually induced a slight increase of the P21 gene expression compared to untreated PE/CA-PJ49 tumor cells. The application of the combined treatment 5 μM CisPt + 40 μM RSV in tumor cells led to a significant increase in P21 gene expression compared to untreated cells (** *p* = 0.002). When using a higher concentration of CisPt together with RSV, a significant increase in P21 gene expression was observed in tumor cells versus control (** *p* = 0.009) or versus RSV (* *p* = 0.046) (Figure 4 and Table 5). These data support the modulatory role of RSV in tumor cells on the effect induced by 20 μM CisPt on the gene expression of P21 without influencing the effect induced by 5 μM CisPt.

Although CRM appears to induce a slight increase in P21 gene on the PE/CA-PJ49 tumor cells, this is not significantly different from P21 expression in untreated cells. The use of combination treatment with 5 μM CisPt + 15 μM CRM showed a significant increase in P21 in tumor cells (** *p* = 0.0014) compared to the effect induced by individual treatment with CRM. Analysis of the induced effect of 20 μM CisPt +15 μM CRM in PE/CA-PJ49 tumor cells also showed a significant increase in P21 gene expression compared to the induced effect of CRM alone (*** *p* = 0.0009) or compared to the 20 μM CisPt effect (** *p* = 0.0018) (Figure 4 and Table 5). These data demonstrate the ability of CRM to support the effect induced by CisPt treatment on P21 gene expression levels.

In conclusion, CisPt treatment led to an increase of the P21 gene expression level of in both lines. RSV treatment for 24 h had a different effect in PE-CA/PJ49 tumor cells (inducing an increase) versus the effect induced in normal HUVEC cells (induced a decrease) in P21 gene expression. CRM alone amplified slightly P21 gene expression in both lines. Applied in combination with CisPt, CRM amplified the effect induced by 5 μM CisPt in both lines, the effect being twice as high in tumor cells. When the high concentration of 20 μM CisPt was used, the effects induced by CRM in tumor cells are the same as those recorded at CRM + 5 μM CisPt. However, in the case of normal cells, 20 μM CisPt + CRM treatment caused a drastic decrease in P21 gene expression, probably due to the toxic effect of CisPt, which killed a large number of normal HUVEC cells.

### 4.5. Effects of the Natural Compounds (RSV,CRM) on Cell Cycle Phases Distribution in Cisplatin Treated Cells

Cell cycle phases distribution was evaluated to further characterise the cytotoxic effect of CisPt and/or RSV, CRM on tumor cells PE-CA/PJ49 compared with HUVEC normal cells.

PE/CA-PJ49 tumor cells and HUVEC normal cells were treated with 5 μM or 20 μM CisPt in the presence or absence of 40 μM RSV or 15 μM CRM for 24 h. The progression through the cell cycle phases induced by treatment were evaluated by flow cytometry using a FACSCanto II flow cytometer. Data analysis were performed using the ModFit 3.2 program which offers the possibility of evaluating the cell cycle phases distribution (G0/G1, S, G2 + M).

The results obtained in the case of the tumor cells PE-CA/PJ49 revealed that single 5 μM or 20 μM CisPt applied treatment decreased of the phase G0/G1 (24.2% or 22.84% versus 41.24 in untreated cells) associated with a slight increase of the synthesis phase (50.5% or 49.6% versus 43.2% in control cells) and followed by the increase of G2+M phase (25.3% or 27.5% versus 15.5% in untreated cells) (Figure 5A,B). The effects induced by the 40 μM RSV treatment in PE/CA-PJ49 tumor cells shown a small decrease of G0/G1 phase (41.24 to 39.05%) associated with an increase in phase S (38.1 to 41.52%) and phase G2 + M (17.56 to 19.41%) compared to untreated cells (Control). Combined CisPt treatment with RSV applied to PE/CA-PJ49 induced an increase of G0/G1 phase (48.4% compared with the untreated cells 41.2%) and a decrease of the synthesis phase (34.1% versus 43.2% in control). Cell cycle phases distribution did not differ significantly when use different concentration of CisPt (5 or 20 μM) and RSV 40 μM are combined. Treatment of PE/CA-PJ49 tumor cells with 15 μM CRM alone or in combination with CisPt induced a decrease in synthesis phase (CRM = 38.6%; CisPt 5 μM + CRM = 33.6%; CisPt 20 μM + CRM = 35.14%; versus control = 43.2%) accompanied by an increase of the G2 +M phase (CRM=33.6%; CisPt 5uM + CRM = 32.2%; CisPt 20uM + CRM = 34.02%; versus control = 15.5%) (Figure 5A,B). These results suggested that the combined CisPt treatment with CRM induced a tendency to block the cell cycle in G2+M phase in tumor cells.

The HUVEC cells used as the normal line showed a distribution in the synthesis phase of less than 20% (untreated cells). The treatments applied individually 5 μM or 20 μM CisPt caused an increase of the synthesis phase to 29.2% and 39.65%, respectively, compared to 19.1% recorded in untreated cells (Control). These data show that treatment with CisPt affects normal HUVEC cells especially at higher concentrations. 40 μM RSV and 15 μM CRM did not appear to significantly affect the cell cycle when applied individually, and cannot influence the effect induced by CisPt treatment when used in combination (Figure 6A,B).

### 4.6. Effects of the Natural Compounds (RSV,CRM) on the Apoptotic Process in Cisplatin-Treated Cells

Apoptosis can be seen as an efficient method of preventing malignant transformation, as it ensures the removal of cells which present with genetic alterations. Deficient apoptosis can promote the development of cancer, both through the accumulation of cells found during division and the blocking of the removal of cells with high malignant potential genetic variations. The factors which determine the cells to follow one of three possible pathways are not yet known – the start of apoptosis after cellular injury, the repair of the lesion or the continuation of the cellular cycle. Many therapeutical agents have anti-tumoral effects generated by their capacity to activate the apoptotic process [68]. Tumoral cells PE/CA-PJ49 and normal cells HUVEC were treated for 24 h with CisPt, RSV, CRM in order to detect the effect induced by treatment with CisPt on tumor cells in the presence or absence of natural compounds (RSV or CRM). The flow cytometric analysis was performed using DIVA 6.2 software and allowed to discriminate viable cells (FITC^−^PI^−^ = Q3) from necrotic cells (FITC^−^PI^+^ = Q1) and early apoptosis (FITC^+^PI^−^ = Q4) from late apoptosis (FITC^+^PI^+^ = Q2).

Analysis the RSV or CRM effects on the apoptotic process of tumor cells PE/CA-PJ49 is shown in the Table 6 and Figure 7. Our results show that the untreated tumor cells PE/CA-PJ49 (Control) had low total apoptosis (Q2 + Q4 = 2%). Treatment with 40 μM RSV induced a raise of total apoptosis to 27.2%, much higher than the total apoptosis induced by 5 μM CisPt (12.5%) or 20 μM CisPt (17.4%). In the case of simultaneous treatment with CisPt and RSV a raise of the total apoptosis was registered (5 μM CisPt + RSV = 31.1%; 20 μM CisPt + RSV = 34.9%) and it seems to have been generated by the presence of RSV.

Treatment for 24 h with 15μMCRM (29.6%) has determined an increase of the total apoptosis in the tumor cells PE/CA-PJ49. Combined treatment CRM with CisPt applied to the PE/CA-PJ49 cells for 24 h has determined a raise of total apoptosis to 28,7% when using 5 μM CisPt or to 40,3% when using 20 μM CisPt (Table 6 and Figure 7).

Effects of the natural compounds RSV or CRM on apoptosis process in normal HUVEC cells were analyzed after applying the same treatment scheme used in the case of PE/CA-PJ49 tumor cells. As shown in Table 7 and Figure 8, treatment with 5 μM or 20 μM CisPt, RSV or CRM did not significantly modify the apoptosis, compared to untreated HUVEC cells (Control). However, combined treatment led to an increase of the apoptotic process. Thus, 5 μM and 20 μM CisPt combined with RSV have determined apoptosis in 14% and 21.6% of the analysed cellular population, a value that was double the one obtained when using individual stimuli (Table 7 and Figure 8). Combined treatment CRM with 5 μM or 20 μM CisPt has induced a raise of apoptosis in 19,7% or 23.9% of the analysed cellular population (Table 7 and Figure 8). Comparative analysis of the obtained data shows that both RSV and CRM have the capacity to stimulate the apoptosis of PE/CA-PJ49 tumor cells without significantly affecting the apoptotic process of normal HUVEC cells (RSV-PE/CA-PJ49 versus RSV HUVEC = 27.2% versus 5.1%; CRM-PE/CA-PJ49 versus CRM-HUVEC = 28.2% versus 7.2%). In conclusion, the two analyzed natural compounds (RSV and CRM) have the ability to amplify the apoptotic process induced by CisPt treatment in PE/CA-PJ49 tumor cells.

## 5. Discussion

The toxic effects of CisPt are dose-dependent and affect the kidneys and bone marrow, which are accompanied by an increase in transaminases and serum creatinine. In addition, CisPt therapy alone has not been found to be effective in treating patients with HNSCC. Cisplatin efficacy appear to be more increased in combination with other chemotherapeutic agents and/or radiation therapy, but side effects or resistance to these drugs are often mentioned [69,70]. This observation suggests that the combination of cisplatin with other anti-tumor agents could be much more effective in the treatment and evolution of head and neck cancer. Some natural compounds due to their chemosensitizing potential, antitumoral activity and ability to reduce the side effects of drugs used in conventional cancer therapy protocols promise to be one of the important components of combinatorial therapy [71,72]. Natural compounds must be well tolerated and have long-lasting benefits. They can also act at the level of different signaling pathways responsible for the tumorigenesis process in HNSCC, which recommends them as agents with multiple molecular targets [73,74]. The experimental approch of this study focused on the enhancing chemotherapy response while lowering side effects and the incidence of drug resistance. To design novel combinatorial therapeutic strategy for improving the patients’ prognosis and response to chemotherapy is needed. To streamline the response to chemotherapy, we analyzed how the use of CisPt in combination with CRM and RSV may influence some cellular processes such as proliferation, P21 gene expression, apoptotic process, and cell cycle development in HNSCC cell line (PE/CA-PJ49) compared to a normal cell line (HUVEC). The results showed that the viability of the tumor and normal cells was affected by CisPt treatment in the same way in both cell lines in a concentration-dependent manner. RSV or CRM treatment affected the viability of tumor cells PE/CA-PJ49 more than of normal HUVEC cells. RSV and CRM have the ability to amplify the inhibitory effect of 5µM CisPt induced on PE-CA/PJ49 tumor cells without significantly affecting the 5 µM CisPt-induced effect in normal cells. The use of a high concentration of CisPt (20 µM) together with a natural compound (CRM or RSV) led to the observation that the effect induced by CisPt on cell proliferation is dominant in both tumor cells and normal cells. Because p21 can act as a tumor suppressor or a tumor-promoting protein [75], and this makes more difficult to establish its function in the evolution of cancer we analysed the role of P21 in response to chemotherapy. Using ELISA and RT-PCR were evaluated protein and gene P21 expression in tumor PE/CA-PJ49 cells and normal HUVEC cells treated with CisPt in presence of RSV or CRM. A significant increase of total p21 protein expression was recorded in PE/CA-PJ49 tumor cells treated with 5 µM CisPt and 20 µM CisPt compared with the effect induced by RSV applied alone. However, it was observed that RSV can potentiate the induced effect of 5 µM CisPt but cannot influence the effects induced by 20µMCisPt treatment on p21 protein expression in tumor cells. In PE/CA-PJ49 the CRM treatment amplified the increase of p21 protein expression induced by CisPt treatment, regardless the CisPt concentration. Increased p21 expression was correlated with a descreased of proliferative activity of tumor cells and a good response to CisPt therapy.

Analysis of P21 gene expression level showed that CisPt treatment induced an increase of P21 expression in both lines. RSV treatment influenced in a different manner P21 expression, inducing an increase in PE/CA-PJ49 and a decrease in normal HUVEC cells. In combination with CisPt, RSV did not influence the CisPt-induced effect in HUVEC cells. On the contrary in PE/CA-PJ49 tumor cells, using the combination CisPt+RSV induced an increase in P21 gene expression and this way, RSV can modulate the effect induced by 20 µM CisPt. CRM alone did not modify the P21 gene expression in both lines. In combination with CisPt, CRM amplified the effect induced by 5 µM CisPt in both lines, the effect being higher in tumor cells. Not significant differences were observed in P21 expression in tumor cells when used the high concentrations of CisPt (20 µM). In normal cells was recorded a drastic decrease in P21 gene expression upon treatment with 20 µM CisPt + CRM. These data support the toxic effect of CisPt used in high concentration on normal cells. The treatment of tumor cells with RSV or CRM in the presence of CisPt induced an increase in protein and gene expression levels of p21. These results may be associated with a more effective response of tumor cells to treatment with lower concentrations of CisPt using RSV or CRM as therapeutic adjuvants.

The cell cycle ensures the cell can proliferate and grow by passing through the G1, S, G2 and M phases. Cancer cells reveal disorders in the cell cycle progression which contributes to exacerbate cell proliferation and to loss of genomic integrity. Treatment with 5 µM or 20 µM CisPt applied alone on tumor cells induced a slight transition from G1 to S phases accompanied by the increase of the G2+M phase (25.3% or 27.5% versus 15.5% in untreated cells). Cell cycle phases distribution did not differ significantly when different concentrations of CisPt (5 or 20 µM) were used. Treatment of PE/CA-PJ49 tumor cells with 40 µM RSV did not affect significantly the cell cycle phases compared to untreated cells. PE/CA-PJ49 cells treated with CisPt + RSV showed an increase of the G0/G1 phase and a decrease in the S phase. Treatment of PE/CA-PJ49 tumor cells with 15 µM CRM alone or in combination with CisPt induced a decrease the number of cells in S phase which was accompanied the massive blockage of cells in G2/M phase. In addition, the results provide evidence that RSV and CRM induce an increase in the apoptotic process of tumor cells. Interaction of RSV or CRM with CisPt in triggering tumor cells apoptosis indicated the amplifier role of the used natural compounds with respect to cisplatin effect. Therefore, associated chemotherapy of CisPt with natural compounds (RSV or CRM) as adjuvants might have a beneficial effect in decreasing the CisPt doses and in reducing its adverse reactions.

## 6. Conclusions

In conclusion, the results suggest that RSV and CRM act against proliferation and sustain the effects of cisplatin by the induction of cell cycle arrest, amplification of DNA damage and contributing to cancer cell destruction by increasing the apoptotic process. In addition, RSV and CRM amplify the expression of P21 in tumor cells and might contribute to increasing the sensitivity to cisplatin.

## Figures and Tables

**Figure 1 nutrients-12-02596-f001:**
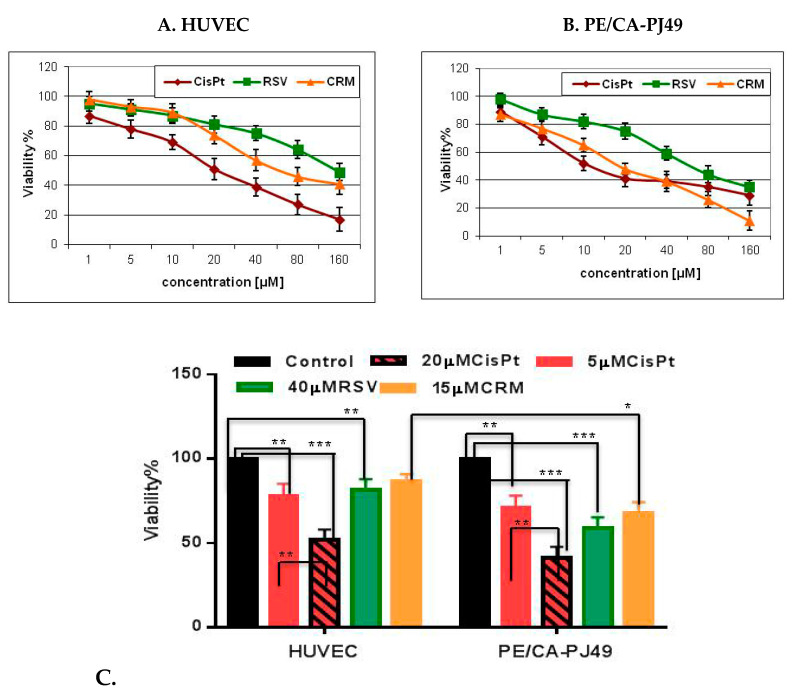
Cisplatin or natural compounds effects on cells viability. (**A**) HUVEC and (**B**) PE/CA-PJ49 cells were either left untreated or treated 24 h with different concentrations of CisPt, CRM. Data shown are representative of three independent experiments and are expressed as mean of three replicates ± SD (*n* = 3). Untreated cells were considered to have 100% viability. Viability% = (T − B)/(U − B) × 100, (where T, absorbance of treated cells; U, absorbance of untreated cells; and B, absorbance of blank). (**C**) Tumor cell viability compared to normal HUVEC cell viability after treatment with CisPt and / or RSV,CRM; (* *p* < 0.05, ** *p* < 0.005; *** *p* < 0.0005).

**Figure 2 nutrients-12-02596-f002:**
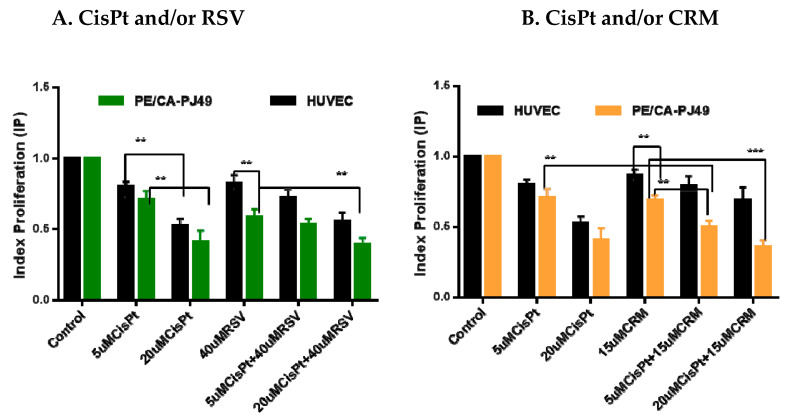
The index proliferation (IP) of HUVEC and PE/CA-PJ49 cells treated with CisPt and/or (**A**) RSV (**B**) CRM was calculated as IP = absorbance of treated cells/absorbance of untreated cells. Results are expressed as DO mean values of three determinations ± standard deviation (SD). Untreated cells were considered to have IP equal 1. (** *p* < 0.005; *** *p* < 0.0005).

**Figure 3 nutrients-12-02596-f003:**
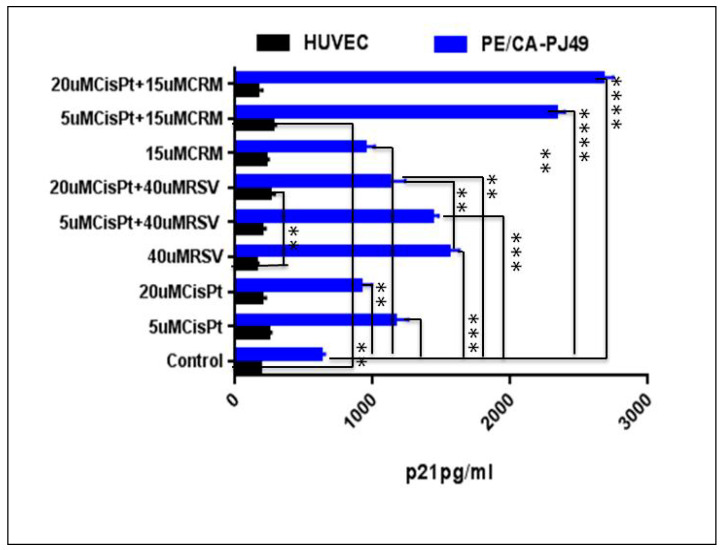
The total p21 protein expression (pg/mL) in normal cell line-HUVEC and in tumor cell line-PE/CA-PJ49 cells treated 24 h with CisPt and/or RSV, CRM. The experiments were performed in triplicates. Results are expressed as mean values of three determinations ± standard deviation (SD). (** *p* < 0.005, *** *p* < 0.0005; **** *p* < 0.00005).

**Figure 4 nutrients-12-02596-f004:**
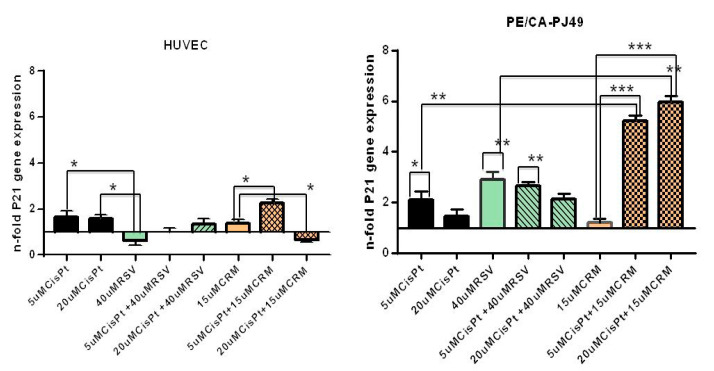
The effect of treatment with CisPt, RSV, CRM applied independently or in combination on P21 gene expression in tumor cells PE/CA-PJ49 compared to normal cells HUVEC. Each sample was performed in duplicate. The samples were analyzed using the formula 2-ΔΔCt = gene expression. (* *p* < 0.05, ** *p* < 0.005; *** *p* < 0.0005).

**Figure 5 nutrients-12-02596-f005:**
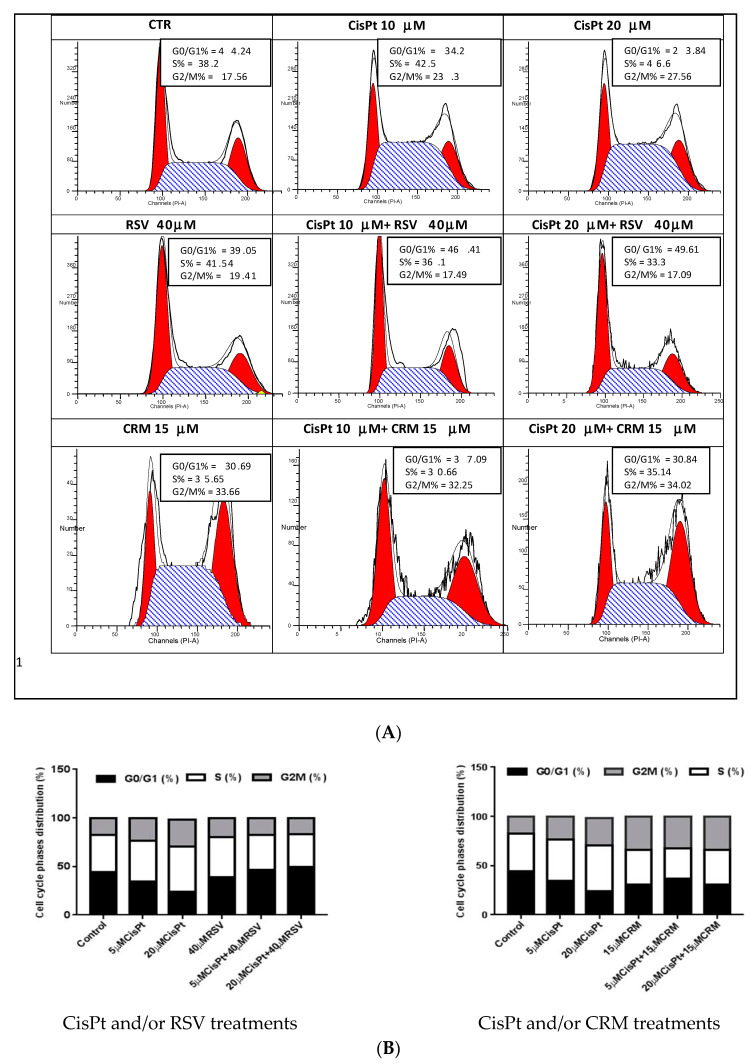
The effect of treatments with CisPt, RSV, CRM applied independently or in combination on tumor cells PE/CA-PJ49 on cell cycle phases distribution. (**A**) flow cytometry histograms; (**B**) cell cycle phases distribution (%).

**Figure 6 nutrients-12-02596-f006:**
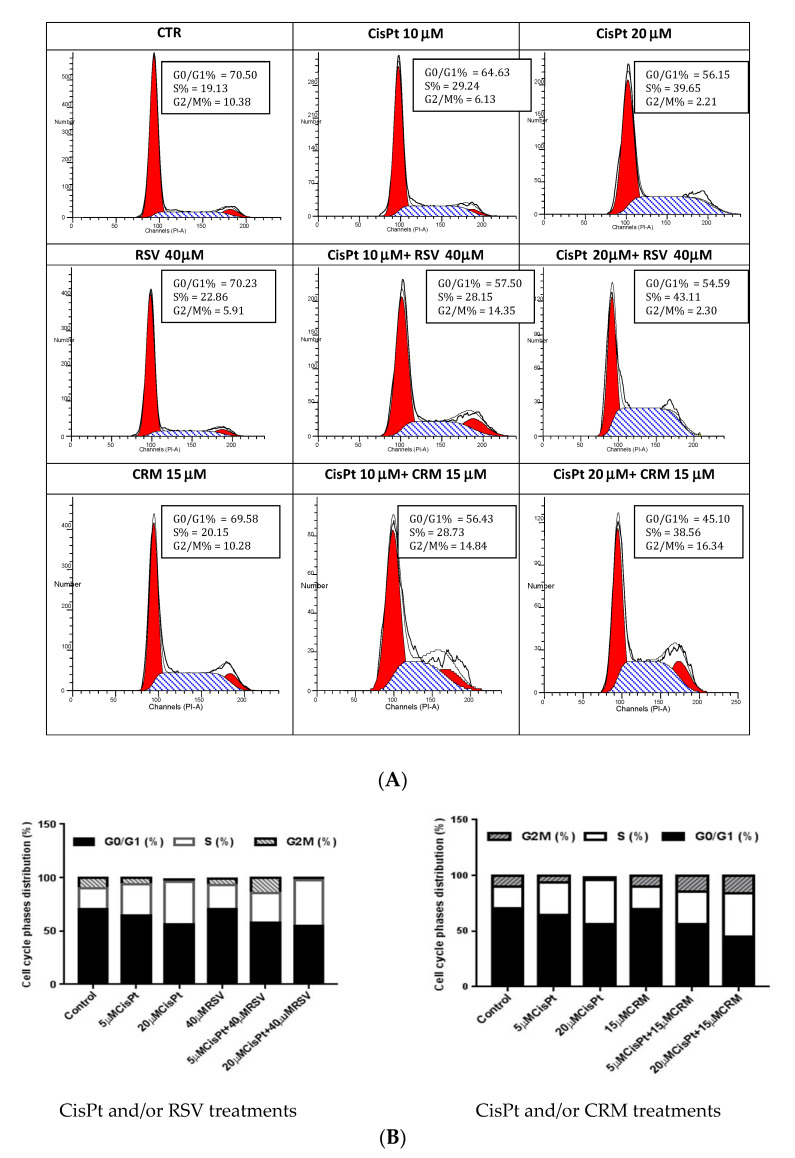
The effect of treatments with CisPt, RSV, CRM applied independently or in combination on normal cells HUVEC on the cell cycle phases distribution. (**A**) flow cytometry histograms; (**B**) cell cycle phases distribution (%).

**Figure 7 nutrients-12-02596-f007:**
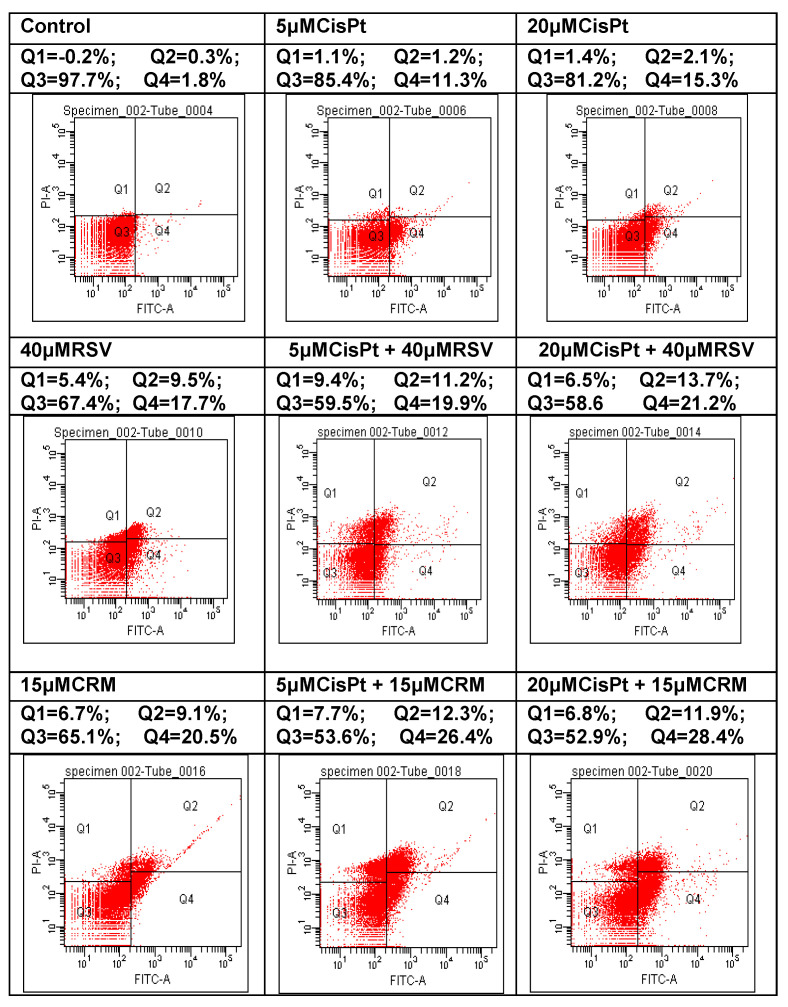
Natural compounds (RSV or CRM) effects on the apoptotic process of PE/CA-PJ49 tumor cells treated 24 h with CisPt and/or RSV or CRM (Dot-plot analysis).

**Figure 8 nutrients-12-02596-f008:**
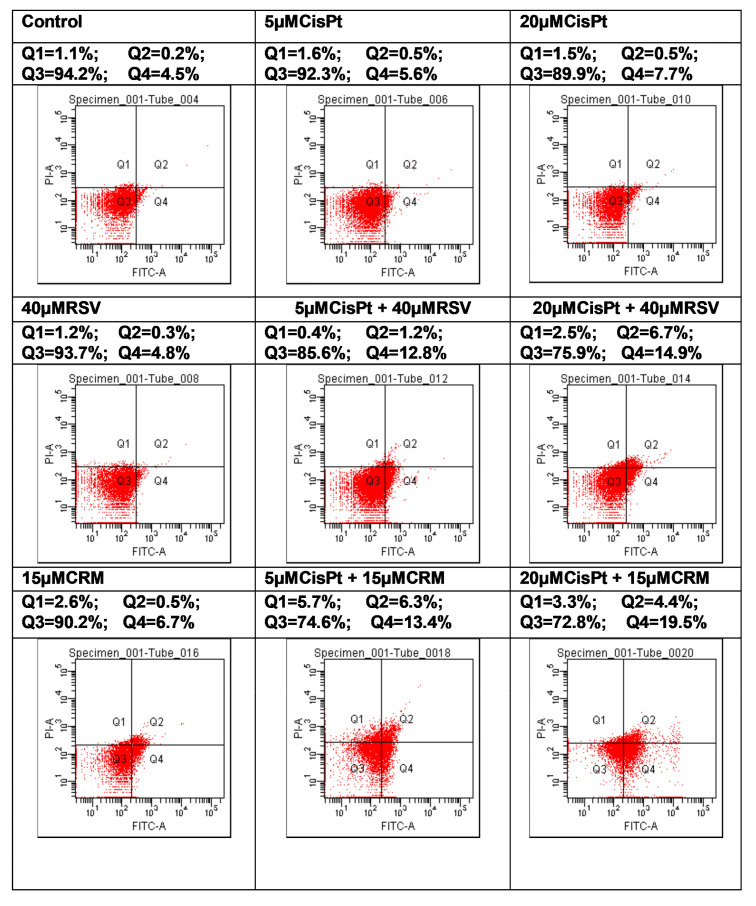
Natural compounds (RSV or CRM) effects on the apoptotic process on normal HUVEC cells treated 24 h with CisPt (Dot-plot analysis.).

**Table 1 nutrients-12-02596-t001:** Preparation of the reaction mixture/reaction.

Reaction Components	Volume
RT-Buffer 10×	2.0 μL
Mix dNTP 25× (100 mM)	0.8 μL
Randomics RT-primeri 10×	2.0 μL
MultiScribe™ Reverse Transcriptase	1.0 μL
Ultrapure water	4.2 μL
Total Volume/reaction	10 μL

**Table 2 nutrients-12-02596-t002:** Real-Time PCR reaction components.

Reaction Components	Volume
QPCR master mix	10 μL
Reference dye	0.3 μL
cDNA	5 μL
Ultrapure water	4.7 μL
Total Volume/reaction	20 μL

**Table 3 nutrients-12-02596-t003:** Inhibitory concentration (IC_25_ and IC_50_) values of the CisPt, RSV and CRM were performed using a linear regression equation for the cytotoxicity curve for PE/CA-PJ49 tumor cells and for normal cells HUVEC. IC_25_ and IC_50_ values are presented as mean ± SEM according to two independent assays, each done in triplicate (**a**). The selectivity index (SI), which indicates the cytotoxic selectivity for CisPt, RSV or CRM against cancer cells versus normal cells, and_._ SI values over 2 were considered as high selectivity (**b**).

(a)						
Treatment (24 h)	HUVEC			PE/CA-PJ49		
	IC 25	IC 50		IC 25		IC 50
CisPt (μM)	5.82 ± 1.1	20.93 ± 2.1		3.95 ± 1.2		9.72 ± 1.7
RSV (μM)	40.6 ± 3.3	110.4 ± 8.6		17.6 ± 1.5		46.8 ± 2.6
CRM (μM)	19.2 ± 2.1	59.3 ± 6.1		7.9 ± 1.8		16.3 ± 3.4
**(b)**						
	**CisPt (μM)**		**RSV (μM)**		**CRM (μM)**	
HUVEC (IC 50)	20.93 ± 2.1		110.4 ± 8.6		59.3 ± 6.1	
PE/CA-PJ49 (IC 50)	9.72 ± 1.7		46.8 ± 2.6		16.3 ± 3.4	
Selectivity Index (SI)	2.15		2.36		3.66	

**Table 4 nutrients-12-02596-t004:** The effects of CisPt, RSV, CRM treatment applied alone or in combination for 24 h on the level of p21 protein expression (n-fold p21protein expression) in normal cell line-Huvec and in tumor cell line-PE/CA-PJ49. The n-fold p21expression was calculated using the formula: n-fold p21expression = p21 pg/mL treatment/p21 pg/mL control.

Treatment	p21 pg/mL HUVEC	n-fold p21 Protein Expression HUVEC	p21 pg/mL PE/CA-PJ49	n-fold p21 Protein Expression PE/CA-PJ49
Control	183	1	625	1
5 μM CisPt	238	1.3	1157	1.85
20 μM CisPt	195	1.07	915	1.46
40 μM RSV	148	0.81	1550	2.48
5 μM CisPt + 40 μM RSV	196	1.07	1429	2.29
20 μM CisPt + 40 μM RSV	253	1.38	1124	1.80
15 μM CRM	221	1.21	941	1.51
5 μM CisPt + 15 μM CRM	272	1.49	2335	3.74
20 μM CisPt + 15 μM CRM	164	0.90	2675	4.28

**Table 5 nutrients-12-02596-t005:** n-fold P21gene expression in PE/CA-PJ49 tumor cells versus normal cells HUVEC treated 24 h with CisPt and/or RSV, CisPt and/or CRM. The results were obtained using gene reference HPRT.

Treatment	HUVEC	PE/CA-PJ49
Control	1	1
5 μM CisPt	1.64	2.0
20 μM CisPt	1.54	1.46
40 μM RSV	0.36	3.1
5 μM CisPt + 40 μM RSV	0.97	2.68
20 μM CisPt + 40 μM RSV	1.34	2.31
15 μM CRM	1.39	1.2
5 μM CisPt + 15 μM CRM	2.26	5.24
20 μM CisPt + 15 μM CRM	0.54	6.03

**Table 6 nutrients-12-02596-t006:** Apoptosis of PE/CA-PJ49 tumor cells induced by 24 h treatment with CisPt and/or RSV, CRM.

Treatment	Necrosis = Q1 (%)	Early Apoptosis = Q4 (%)	Late Apoptosis = Q2 (%)	Total Apoptosis = Q2 + Q4 (%)
Control	0.2	1.8	0.3	2.1
5 μM CisPt	2.1	11.3	1.2	12.5
20 μM CisPt	1.2	15.3	2.1	17.4
40 μM RSV	5.4	17.7	9.5	27.2
5 μM CisPt + 40 μM RSV	9.4	19.9	11.2	31.1
20 μM CisPt + 40 μM RSV	6.5	21.2	13.7	34.9
15 μM CRM	6.7	19.1	9.1	28.2
5 μM CisPt + 15 μM CRM	7.7	26.4	12.3	38.7
20 μM CisPt + 15 μM CRM	6.8	28.4	11.9	40.3

**Table 7 nutrients-12-02596-t007:** Apoptosis of HUVEC cells induced by 24 h treatment with CisPt and/or RSV, CRM.

Treatment	Necrosis = Q1 (%)	Early Apoptosis = Q4 (%)	Apoptoza Tarzie = Q2 (%)	Apoptoza Totală = Q2+Q4 (%)
Control	1.1	4.5	0.2	4.7
5 μM CisPt	1.6	5.6	0.5	6.1
20 μM CisPt	1.5	7.7	0.5	8.2
40 μM RSV	1.2	4.8	0.3	5.1
5 μM CisPt + 40 μM RSV	0.4	12.8	1.2	14
20 μM CisPt + 40 μM RSV	2.5	14.9	6.7	21.6
15 μM CRM	2.6	6.7	0.5	7.2
5 μM CisPt + 15 μM CRM	5.7	13.4	6.3	19.7
20 μM CisPt + 15 μM CRM	3.3	19.5	4.4	23.9

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
