# Peer review of "The Effect of Resveratrol or Curcumin on Head and Neck Cancer Cells Sensitivity to the Cytotoxic Effects of Cisplatin"

_nutrients, 2020, doi:10.3390/nu12092596_

Round 1

Reviewer 1 Report

The present manuscript deals with the influence of the natural products curcumin and resveratrol on p21 protein or gene expression in head and neck squamous cell carcinomas (HNSCC) line (PE/CA-PJ49) with or without cisplatin treatment, and its comparison with the control Human umbilical vein endothelial cell line (HUVEC). The authors also examined the influence of both natural products on the apoptotic processes, the tumor cell arrest, and the proliferation activity of tumor cells against cisplatin therapy. The authors investigated the possible correlation between p21 expression and the apoptotic processes or cell cycle progression in PE/CA-PJ49 tumor cells compared to HUVEC control cells. Showing the potential effects of curcumin and resveratrol as adjuvants in cisplatin therapy allowing the decreased of cisplatin dosage, and thus reducing the adverse effects of cisplatin.

The manuscript is easy to read and understand. The experiments carried out, namely, MTT assay, cell proliferation assay, BCA assay, ELISA assay for P21 protein quantification, flow cytometry for apoptosis and cell cycle analysis, and Real-Time PCR   to analyze the CDKN1A gene (P21) expression level fitted the aims of the manuscript.

Although, curcumin and resveratrol have been already studied as therapeutic agents for head and neck carcinomas therapy, the findings of the present manuscript are interesting for a broader community. Even though, the paper comes with a few issues, which are addressed below:

Revise the figure 1, the panels lack letter, the figure caption does not reflect the third panel; use different colors or shades for the two cisplatin concentrations displayed.

Revise the MTT name as the hyphens are missing [3 (4,5 dimethylthiazol 2 yl) 2,5 di¬phenyltetrazolium bromide]à MTT 3-(4,5-dimethylthiazol-2-yl)-2,5-diphenyltetrazolium bromide

Revise the space between number and units throughout the text to make it more coherent:

  • Line 155: maintained at 37  C vs Line 167: 37°C vs Line 240: 370C vs Line 262: 4 ° C
  • Line 183: 1-80 μM vs Line 229: 400μl of vs Line 273: 20ul

There are some typos along the manuscript:

  • Tables 1 and 2 Volum à volume
  • Line 281: 999ΔCt1 à ΔCt1
  • Line 300: Tabel à Table
  • Table 4: tratament à treatment
  • Line 462: Semnificative à significative

Author Response

Response to Reviewer 1

We are very grateful for the reviews provided by you as an external reviewer of this manuscript. The revised manuscript takes into account these aspects.

Please see below in red our response to your comments (highlighted in the revised manuscript).

Point 1: Revise the figure 1, the panels lack letter, the figure caption does not reflect the third panel; use different colors or shades for the two cisplatin concentrations displayed.

  • Response 1

Figure 1: the corrections are made; we have revised the figure following your comments see L 338-339

Point 2: Revise the MTT name as the hyphens are missing [3 (4,5 dimethylthiazol 2 yl) 2,5 di¬phenyltetrazolium bromide]à MTT 3-(4,5-dimethylthiazol-2-yl)-2,5-diphenyltetrazolium bromide

  • Response 2: The correction is made, see L 175

Point 3: Line 155: maintained at 37  C vs Line 167: 37°C vs Line 240: 370C vs Line 262: 4 ° C; Line 183: 1-80 μM vs Line 229: 400μl of vs Line 273: 20ul

  • Response 3: We have revised accordingly

Point 4: There are some typos along with the manuscript:

Tables 1 and 2 Volum à volume

Line 281: 999ΔCt1 à ΔCt1

Line 300: Tabel à Table

Table 4: tratament à treatment

Line 462: Semnificative à significative

  • Response 4: We have revised accordingly

Sincerely yours,

Viviana Roman

Reviewer 2 Report

The manuscript describes the modulation of the anticancer potential of platinum-based drugs (CisPt) caused by resveratrol and curcumin in in vitro conditions. The experimental protocol was well structured and the experiments well conducted.

Moreover, some changes throughout the text are needed. 

My suggestions to improve the manuscript:

First of all, the title does not reflect the content of the paper. I strongly suggest changing the title e.g. The effect of resveratrol and curcumin on head and neck cancer (HNSCC) cells sensitivity to the cytotoxic effects of CisPt in in vitro conditions”

The work is of limited relevance to in vivo conditions. Using the term “chemotherapy” and referring to clinical applications is exaggerated, e.g.  L. 29-30 & L. 66-69 & L. 126-129 - exaggerated claims, sentences should be removed; 

The introduction is unusually long and may be shortened. The aim of the study may be more briefly described. 

The units are missing several times in the text (e.g., L. 302-305) and in Tables.

Abbreviations used in all Tables and Figures should be explained in the legends. 

Commas in Table 3 and 7. 

Captions of Figure 1, 5 and 6 should be improved. 

Quality of Figure 5 and 6 should be improved.

Table 7. Content of table 7 should be checked.

  1. 70 – It should be Curcuma longa L.  
  2. 78 – RSV
  3. 79-81 – Please, add references

Author Response

Response reviewer 2

            We are very grateful for the reviews provided by you as an external reviewer of this manuscript. The revised manuscript takes into account these aspects.

Please see below in red our response to your comments (highlighted in the revised manuscript).

Point 1: First of all, the title does not reflect the content of the paper. I strongly suggest changing the title e.g. The effect of resveratrol and curcumin on head and neck cancer (HNSCC) cells sensitivity to the cytotoxic effects of CisPt in in vitro conditions”

  • Response 1: Thank you for your suggestion; we change the title

The effect of resveratrol or curcumin on head and neck cancer cells sensitivity to the cytotoxic effects of cisplatin

Point 2: Using the term “chemotherapy” and referring to clinical applications is exaggerated, e.g.  L. 29-30 & L. 66-69 & L. 126-129 - exaggerated claims, sentences should be removed; 

  • Response 2: This sentence is revised

L 29-30 In conclusion, using RSV or CRM as adjuvants in CisPt therapy might have a beneficial effect in decreasing the CisPt doses and in reducing the adverse reactions in patients with HNSCC.

In conclusion, using RSV or CRM as adjuvants in CisPt therapy might have a beneficial effect on supporting the induced effects of CisPt .

L66-69: Sentence is removed

In this study we have analyzed how chemotherapy may be improved, in order to achieve a good prognosis while minimising adverse reactions of the current treatment of head and neck cancer. We selected two natural compounds, such as curcumin and resveratrol, which are known to have anticancer properties and could be used as adjuvants in cancer management

L126-129: This sentence is revised

The use of natural compounds CRM or RSV, as adjuvants in CisPt therapy might have a beneficial effect in decreasing the CisPt doses and in reducing the adverse reactions induced by a chemotherapeutic agent.

The results of this study suggest the use of natural compounds CRM or RSV, as adjuvants in order to improve the response to CisPt therapy.

Point 3: The units are missing several times in the text (e.g., L. 302-305) and in Tables.

  • Response 3: We have revised accordingly L297-325

The IC50 values (±SEM) for CisPt were 9.72+/-1.7μM on PE/CA‑PJ49 cells and 20.93 +/-2.1 μM on HUVEC cells (Table 3A). RSV, under our experimental conditions, had an IC50 of 46.8±2.6 μM on PE/CA‑PJ49 cells and 110.4+/-8.6 μM on HUVEC cells (Table 3). CRM treatment for 24h reduced the cellular viability to an IC50=16.3+/-3.4 μM on PE/CA‑PJ49 tumor cells and IC50=59.3+/-6.1 μM on normal cells HUVEC (Table 3A).

Point 4: Commas in Table 3 and 7. 

  • Response 4: We have revised accordingly

Table 3. IC25 and IC50 (inhibitory concentration) values of the CisPt, RSV, and CRM were performed using a linear regression equation for the cytotoxicity curve for PE/CA‑PJ49 tumor cells and for normal cells HUVEC. IC25 and IC50 values are presented as mean ± SEM according to two independent assays, each done in triplicate (A). The selectivity index (SI), which indicates the cytotoxic selectivity for CisPt, RSV or CRM against cancer cells versus normal cells, and. SI values over 2 were considered as high selectivity (B).

A.

Treatment

(24h)

HUVEC

PE/CA‑PJ49

IC 25

IC 50

IC 25

IC 50

CisPt (μM)

5.82+/-1.1

20.93 +/-2.1

3.95+/-1.2

9.72+/-1.7

RSV (μM)

40.6+/-3.3

110.4+/-8.6

17.6+/-1.5

46.8+/-2.6

CRM (μM)

19.2+/-2.1

59.3+/-6.1

7.9+/-1.8

16.3+/-3.4

B.

CisPt (μM)

RSV (μM)

CRM (μM)

HUVEC (IC 50)

20.93 +/-2.1

110.4+/-8.6

59.3+/-6.1

PE/CA‑PJ49 (IC 50)

9.72+/-1.7

46.8+/-2.6

16.3+/-3.4

Selectivity Index (SI)

2.15

2.36

3.66

Table 7. Apoptosis of HUVEC cells induced by 24h treatment with CisPt and / or RSV, CRM.

Treatment

Necrosis=

Q1 (%)

Early apoptosis=Q4(%)

Apoptoza tarzie=

Q2 (%)

Apoptoza totală=

Q2+Q4 (%)

Control

1.1

4.5

0.2

4.7

5μM CisPt

1.6

5.6

0.5

6.1

20μM CisPt

1.5

7.7

0.5

8.2

40μM RSV

1.2

4.8

0.3

5.1

5μM CisPt + 40μM RSV

0.4

12.8

1.2

14

20μM CisPt + 40μM RSV

2.5

14.9

6.7

21.6

15 μM CRM

2.6

6.7

0.5

7.2

5μMCisPt + 15μM CRM

5.7

13.4

6.3

19.7

20μM CisPt + 15μM CRM

3.3

19.5

4.4

23.9

Point 5: Captions of Figure 1, 5 and 6 should be improved. Quality of Figure 5 and 6 should be improved

  • Response 5: We have revised accordingly

L 340 - Figure 1

L 529 - Figure 5

L 542- Figure 6

Sincerely yours,

Viviana Roman